# Modeling Liver Development and Disease in a Dish

**DOI:** 10.3390/ijms242115921

**Published:** 2023-11-02

**Authors:** Waqas Iqbal, Yaru Wang, Pingnan Sun, Xiaoling Zhou

**Affiliations:** 1Stem Cell Research Center, Shantou University Medical College, Shantou 515041, China; waqasbiotech@yahoo.com (W.I.); 22yrwang@stu.edu.cn (Y.W.); pnsun@stu.edu.cn (P.S.); 2Research Center for Reproductive Medicine, Shantou University Medical College, Shantou 515041, China; 3Guangdong Provincial Key Laboratory of Infectious Diseases and Molecular Immunopathology, Shantou University Medical College, Shantou 515041, China

**Keywords:** human pluripotent stem cell, liver development, organoids, metabolic liver disease, viral infection, primary liver cancer, regenerative medicine, organ-on-chip

## Abstract

Historically, biological research has relied primarily on animal models. While this led to the understanding of numerous human biological processes, inherent species-specific differences make it difficult to answer certain liver-related developmental and disease-specific questions. The advent of 3D organoid models that are either derived from pluripotent stem cells or generated from healthy or diseased tissue-derived stem cells have made it possible to recapitulate the biological aspects of human organs. Organoid technology has been instrumental in understanding the disease mechanism and complements animal models. This review underscores the advances in organoid technology and specifically how liver organoids are used to better understand human-specific biological processes in development and disease. We also discuss advances made in the application of organoid models in drug screening and personalized medicine.

## 1. Introduction—Historical Overview

Stem cells have revolutionized how we perceive developmental biology, disease modeling, and the development of possible therapies. However, the main focus has been to obtain a considerably pure population of a specific cell type [1]. However, the advent of organoid technology (Figure 1), a three-dimensional (3D) culture system that accurately emulates an organ, has opened new avenues in the field of cell culture [2,3,4]. 

Organoids possess key characteristics of organs, such as two or more types of cells of the organ it mimics, displaying similarity in function and resembling cellular organization. Hence, an organoid can be defined as a miniature organ containing several types of cells obtained via the differentiation of stem cells or progenitor cells that self-organize and embody lineage commitment [15]. So far, organoids have been developed for organs originating from all three germ layers (Figure 2) and are being used as model systems in basic and translational research.

The advancement in the organoid technology has its roots in the fields of stem cell biology, extracellular matrix (ECM) biology, and the reorganization of tissues dissociated ex vivo [23]. Bissel and colleagues [24] observed improved hepatocellular function in rat hepatocytes in the presence of ECM. Furthermore, mammary gland epithelia were able to develop tubules and ducts when embedded in hydrogels containing ECM [24,25]. Interestingly, coculturing gastric epithelia and fibroblast in hydrogels in the presence of ECM enabled a successful, albeit short-term, culture of stomach tissues and their differentiation into gastric mucosal cells [26]. Although these cultures had hallmarks of organoids and signified the importance of 3D culture systems, they were not self-renewable; hence, sustained long-term cultures were not possible. However, these shortcomings were overcome by culturing leucine-rich repeat-containing G-protein-coupled receptor 5 (Lgr5)-positive adult stem cells isolated from mouse intestinal tissue. These cells were able to self-renew and demonstrate morphological and functional characteristics of intestinal crypt-villus [27]. The addition of the LGR ligand, R-spondin, in the culture media mediated the up-regulation of the Wnt signaling pathway, thus facilitating stem cell maintenance and organoid formation from *Lgr5^+^* intestinal progenitor cells [28]. Further optimization of the R-spondin-containing media led to the generation of organoids from lgr5^+^ progenitor cells from the colon [28], stomach [29], and liver [30]. In addition to the progenitor cells, embryonic stem cells (ESCs) have also been shown to differentiate into cortical tissues in a 3D culture [31]. This has led to the generation of cerebral organoid models that recapitulate the developmental aspects of the human brain [32]. Subsequent studies have successfully generated organoids using PSCs and tissue-specific progenitor cells to model organs derived from the ectoderm, mesoderm, and endoderm [33] (Figure 2), including the liver [12,30,34,35] (Table 1).

Ever since the human embryonic stem cells (hESCs) and induced pluripotent stem cells (iPSCs) were established [36,37,38], numerous cell types have been generated via differentiation protocols recapitulating the essence of organogenesis in vivo [39] (Table 1). The establishment of organoids over the years has added a new paradigm to modelling human diseases (Table 2). Organoids can be defined as three-dimensional structures that originate from stem/progenitor or differentiated cells and self-organize via cell-to-cell and cell-to-matrix interactions, thus recapitulating the tissue-specific structure and function in vitro [40] in a way that the traditional cell cultures and animal models lack. In this review, we discuss the importance of liver organoids and the generation of organoids from hPSCs and tissue-resident stem cells. We also discuss how organoids can further our understanding of the mechanisms involved in disease progression and provide opportunities in developing personalized therapies.
ijms-24-15921-t001_Table 1Table 1A general overview of the organoid technology in disease modeling for different organs.
Organ

Organoid

Model System

Cell Source

Media and Supplements

Salient Features

Species

Refs.
**
Brain
**Cerebral organoidMicrocephalyhPSCs/mouse ES cellsD0; human ES media, bFGF, and ROCK inhibitor. D6; DMEM/F12, N2, Glutamax. MEM-NEAA and heparin. D11; DMEM/F12 and Neurobasal containing N2, B27 without vitamin A, 2-mercaptoethanol, insulin, Glutamax and MEM-NEAA. D15; DMEM/F12 and Neurobasal containing N2, B27 with vitamin A, 2-mercaptoethanol, insulin, Glutamax and MEM-NEAACerebral organoids that recapitulate cortical developmentHuman/Mouse[32]**
Lungs
**Organoids consisting of epithelia and alveoli-like structuresFetal lung hPSCsD0; RPMI 1640, Activin A, and dFBS. D4; anterior foregut, Advanced DMEM/F12, N-2 and B27, Hepes, L-Glutamine, Penicillin–streptomycin, Noggin, SB431542 and CHIR99021Development of fetal-like lung organoids with proximal airway structuresHuman[41]**
Liver
**Functional liver budLiver failure modeliPSCs, MSCs, HUVECsD0; RPMI 1640, B27 and Activin A. D6; RPMI1640, B27, human bFGF, human BMP4iPSCs-derived liver bud transplantation in liver failure modelHuman[42]**
Stomach
**Gastric organoidsPyloric epithelial organoidsASCsGastric culture medium; Advanced DMEM/F12, B27, N2, nAcetylcysteine, and Gastrin containing EGF, R-spondin1, Noggin, FGF10 and Wnt3A. Enteroendocrine lineage; Exendin 4Lgr5^+^ stem cells at the base of pyloric glands drive epithelial regenerationMouse[29]**
Kidney
**Kidney organoidsFetal kidney organoidiPSCsD0; APEL basal medium, CHIR99021 and Antibiotic–Antimycotic. D5; APEL basal medium, FGF9 and heparin. D7; APEL and CHIR99021 (1h), APEL, FGF9 and heparin. D13; APEL basal medium Generation of multicellular fetal-like kidney organoids using small moleculesHuman[43]**
Pancreas
**Pancreatic organoids consisting of ductal and acinar progenyCystic fibrosis modeliPSCsD0; BE1 (MCDB13, glucose, sodium bicarbonate, fatty acid free, BSA, L-glutamine) supplemented with CHIR99021 and Activin A. D1; BE1 with Activin A. D4; BE1 with KFG. D6; BE3 (MCDB131 with 0.44 g/L glucose, 1.754 g/L sodium bicarbonate, FAF-BSA, L-glutamine, L-ascorbic acid, ITS-X) with SANT-1, retinoic acid, LDN-193189 and PD0325901. D10; BE3 with FGF10, Indolactam V, SB431542 and glucose. D12; BE3 with Rock inhibitor. D14; BE3, FGF2 and Rock inhibitor. D18; BE3 with FGF2 and nicotinamidePatient-derived pancreatic organoids as in vitro cystic fibrosis modelHuman[44]**
Intestine
**Intestinal organoidsCrypt-villus organoids recapitulating intestinal cryptsASCsCulture media; Advanced DMEM/F12, EGF, R-spondin 1, Noggin and Jagged-1 peptide. Overlay media; Advanced DMEM/F12, EGF, R-spondin 1, Noggin and Y-27632Lgr5+ cells can form crypt-villus organoids in vitroMouse[27]Organoids have been generated for a number of organs as depicted in the table using pluripotent stem cells and tissue-resident stem cells. Direct differentiation of PSC-derived organoids require the induction of germ-layer specification following induction of specific cell types and finally maturation of cells via the introduction of growth factors. Organoids derived from adult stem cells require the isolation of tissue-specific stem cells. iPSCs, induced pluripotent stem cells; ES, embryonic stem cells; ASCs, adult stem cells; MSCs, mesenchymal stem cells; hPSCs, human pluripotent stem cells; HUVECs, human umbilical vein endothelial cells. bFGF, basic fibroblast growth factor; ROCK, Rho-associated protein kinase; DMEM/F12, Dulbecco’s modified eagle medium/F12; MEM-NEAA, minimum essential media–nonessential amino acids; dFBS, defined fetal bovine serum; FGF, fibroblast growth factor; BMP, bone morphogenetic protein; EGF, epidermal growth factor; KFG, keratinocyte growth factor; FAF-BSA, fatty acid-free bovine serum albumin; ITS-X, insulin–transferrin–selenium–ethanolamine.
ijms-24-15921-t002_Table 2Table 2Organoids and organ-on-chip in modeling liver diseases.
Organ

Disease Category

Disease Modelled

Platform

Cell Source

Cell Repertoire in Organoid

Study Objective

Advantages

Constraints

Refs.
**
Liver
**Metabolic diseasesNAFLD/MASLDOrganoidESCs/iPSCs  (Human)|Fetal hepatocytesHepatocytes, Stellate cells, and Kupffer cells|HepatocytesInvestigate steatohepatitisPatient-derived organoids in precision medicine|Precise modelingBatch to batch variability|CRISPR-based gene editing[18,45]Organ-on-chipHepatocytes, Stellate cells, Kupffer cells, Sinusoidal cells  (Human primary)Hepatocytes, Stellate cells, Kupffer cells, Sinusoidal endothelial cellsNAFLD progression to NASHNon-invasive model consisting of all the major cells involved in NASH progression as a preclinical systemMaintain dynamic flow conditions. [46]ALDOrganoidESCs, hFLMCs  (Human)Hepatocytes, CholangiocytesDevelopment of alcoholic liver injury model that improved survivability of FRG mice after transplantationDeveloped robust functional and transplantable organoidsFetal-like characteristics[47]Organ-on-chipPHH, LSECs, Kupffer cells  (Human)Hepatocytes,  Sinusoidal endothelial cells, and Kupffer cellsAlcoholic liver disease model that recapitulates blood alcohol concentrationsPrimary cells recapitulating alcoholic liver disease in a triculture systemStudy lacks long-term culture[48]Viral infectionsSARS-CoV-2OrganoidProgenitor cells  (Human)Human ductal cellsSARS-CoV-2 ductal organoid modelTo test antiviralsLow infection rate[13]Organ-on-chip------HEVOrganoidBipotent progenitor cells  (Human)Hepatocytes and CholangiocytesDevelopment of experimental model for hepatotropic virus (HEV) infectionIdentification of antiviral drugsLiver biopsies[49]Organ-on-chip------HBVOrganoidiPSCs, HUVECs, and BM-MSCs  (Human) 
Hepatocytes, Mesenchymal stem cells, and endothelial cellsDevelopment of patient-derived organoid model system for HBV infection Compared to 2D culture systems the organoids were more prone to HBV infectionLack of adult phenotype[50]Organ-on-chipPHH and KC  (Human)Primary hepatocytes and Kupffer cellsMicrofluidic system that recapitulates HBV life cycle and innate immunityThe system provides an in-depth analysis of HBV and human immune response Activation of Kupffer cells is required via exogenous stimulant [51]HCVOrganoid------Organ-on-chipHepatoblasts and CD8^+^ T cells  (Human)Hepatocytes and T CellsMicrofluidic model to study adaptive immune response against HCVAn alternative to surrogate animal modelsLack of other immune cells. Extracellular matrix impedes T Cell and organoid contact[52]Primary Liver CancerCCOrganoidTumor cellsCholangiocarcinoma cellsBiomarker identification and drug screeningPersonalized drug testingLack of immune and stromal cells [53]Organ-on-chip------HCCOrganoidTumor cellsPrimary liver cancer cellsBiomarker identification and drug screeningPatient-derived organoids for personalized medicineModel deficit in immune cells[53]Organ-on-chip------CHCOrganoidTumor cellsCholangiocarcinoma cells and Hepatocellular carcinoma cellsBiomarker identification and drug screeningIn vivo model for drug screening that mimics tumor architecture and gene expressionLack of stromal and immune cells[53]Organ-on-chip------Liver diseases modeled using organoids and organ-on-chip. Liver organoids and organ-on-chips have been successfully used to model liver diseases, identify biomarkers, and further our understanding of personalized medicine. NAFLD, non-alcoholic fatty liver disease; MASLD, Metabolic dysfunction-associated steatotic liver disease; ALD, alcoholic liver disease; SARS-CoV-2, severe acute respiratory syndrome coronavirus disease 2; HEV, Hepatitis E virus; HBV, Hepatitis B virus; HCV, Hepatitis C virus; CC, Cholangiocarcinoma; HCC, Hepatocellular carcinoma; CHC, combined hepatocellular–cholangiocarcinoma.

## 2. Animal Models and Divergences in Developmental Trajectories

Many aspects of biological principles are evolutionarily conserved across species; this enables animal models, such as *C. elegans*, *D. melanogaster*, and *Mus musculus*, which is resourceful in understanding developmental processes and disease [54]. Mice have been especially useful as disease-related models due to their physiological relevance, availability of genetic tools, small size, and rapid reproduction. Nevertheless, fundamental differences exist between species which necessitates the need for human-specific model systems in understanding human development and disease. The need for such models is ever so important in understanding the genetic variability and susceptibility to genetic diseases; for instance, neurological and neurodevelopmental disorders, and devising therapeutics. Developmental processes such as embryonic and fetal development are species-specific and differ greatly from species to species (Figure 2) with different life spans. The complex nature of the human brain is considered to be a main factor for its prolonged developmental time scale [55,56], as evident by the fact that the synapse formation and dendritic development takes place from months to years in humans. Moreover, the process of synaptic elimination continues over decades in humans, whereas cortical neurons reach maturity after 5 weeks in the mouse brain and 4 months in macaques [55].

Apart from the complexity and slower pace of the developmental process of the human brain, other aspects of human development occur at a slower pace as well. Somite development in vertebrates is controlled via the oscillation of a gene expression pattern termed as the “somite segmentation clock”, which lasts 5–6 h in humans as compared to 2–3 h in mouse [57]. Likewise, the motor neurons take 2.5 times longer to develop in humans in comparison to mice [58], which is attributed to increased protein stability and cell cycle duration in human cells [57,58]. This underlies the disparity in metabolic processes that could influence the developmental time scale in species.

The prolonged time scale for development in humans is also attributed to diversity in progeny cells [59], as evident by the evolution of the human neocortex [60], especially in the supragranular layers, which shows increased diversification in glutamatergic neurons compare to the mouse neocortex [61]. The highly divergent neuronal subtypes, especially in cortical layer 3, could render increased susceptibility to Alzheimer’s disease [62], consequently making species-specific disparities ever so important in the analysis of human diseases.

Animal models have been extensively used to help determine pharmacological and biological actions of drugs. However, in the past 20 years, numerous candidate drugs showing encouraging results in preclinical models of liver diseases, such as NASH, failed during clinical trials [63,64]. Additionally, there seems to be a lack of consensus on which NAFLD model recapitulates human liver disease more closely [65]. Furthermore, genetic models of human diseases fail to recapitulate the mechanisms found in patients [66]. The inconsistency in study design and interlaboratory variability severely limits the reproducibility of in vivo experiments [67,68].

Inter-species differences also dictate susceptibility to pathogenic diseases. For instance, *Helicobacter pylori*, which causes gastritis and is a known factor for causing adenocarcinoma in humans [69], can infect mice but does not cause adenocarcinoma in wild-type mice, thus limiting their use as animal models. Viruses are also known to be host-specific. Severe acute respiratory syndrome coronavirus 2 (SARS-CoV-2), which causes COVID-19 in humans (Figure 3), does not infect wild-type mice [70]. Mosquito-borne Zika virus, which causes birth defects and microcephaly in human fetuses, only causes microcephaly in the offspring of wild-type mice when the virus is injected in the placenta or fetal brain and not when the pregnant mice are injected with the virus subcutaneously [71,72]. Lack of a proper murine infection model impedes the in vivo assessment of HBV drugs [73]. The expression of hNTCP in the murine model helps facilitate viral entry but fails to establish viral infection [74]. On the other hand, mice are resistant to HCV infection and several transgenic mouse lines have to be developed to study different aspects of HCV infection [75]. Consequently, this necessitates the use of organoid models to overcome the limiting factors that species-specific differences pose, such as developmental timing, diversity in cell types, and genetic predisposition.

## 3. Organoid Technology: The Inception of an Organ in a Dish

Understanding how ECM can impact cellular organization and behavior has been instrumental in discovering the self-organizing potential of stem cells. This is evident by alveolar structures formed by mammary epithelia in the presence of a reconstituted basement membrane [77,78]. In addition to their structural resemblance, epithelial polarization, and compartmentalization to alveoli, they remarkably emulated alveolar function, evident by the tissue-specific vectorial secretion of milk into the lumina. Although these alveolar models recapitulating the alveolar architecture could be considered as early examples of organoids, it took over two decades for the field of organoids as seen today, when a single Lgr5^+^ adult stem cell (ASC) was shown to give rise to miniature tissue that resembled the stomach and small intestine [27,29]. Advancement in the field of stem cell biology has led to many pioneering work using embryonic stem cells (ESCs) in developing organoids for the brain [31], optic cup [79], and pituitary gland [80]. The field of organoids really took pace when induced-pluripotent stem cells (iPSCs) and adult stem cell-derived organoids were utilized as model systems for human diseases [32,81]. Numerous organoid models have been developed since then for human organs. However, PSC- (pluripotent stem cells) and ASC-derived organoids are used in different context. Since ESCs/iPSCs give rise to all the three germ layers, PSC-derived organoids are considered developmental models. The organoid cultures follow the developmental process that an embryo goes through, such as the induction of the endoderm, ectoderm, or mesoderm mimicking embryonic and fetal development. This allows us to study the human developmental process to answer questions that have been eluding us thus far. However, PSCs fail to emulate the adult stages of tissue development, thus they can be deemed more suitable for understanding embryonic and fetal development and diseases. Organoids derived from ESCs and iPSCs show remarkable histological similarities to the developmental stages of an organ they represent [54]. Multiple developmental stages of an organ in vivo are emulated in a step-by-step protocol for the development of PSC-derived organoids [31,32,54,82]. Generally, the differentiation process starts with the expansion of PSCs and the formation of embryoid bodies [31,32,82]. This is followed by the addition of exogenous morphogens to simulate the developmental process step by step (Figure 2). For instance, the development of brain organoids starts with the formation of embryoid bodies that are introduced to a morphogen-containing medium for neuronal differentiation [31,32,83,84]. Once the fate of the cells is decided and the cells in the progenitor state proliferate, the medium is switched again to promote maturation of the neurons [83,85]. Consequently, the stage wise exposure of organoids to different conditions in vitro recapitulates the development stages of human fetal organ developmental.

How cells interact with the surrounding ECM is instrumental in the establishment of organoids. ECM for organoid cultures is generally animal-based and derived from ECM-producing tumors providing structural support in addition to growth factors and cell-matrix signals for the optimal development of organoids. There are different ways of administering ECM components; dissolve the matrix in the medium used for the organoid culture or embed the organoids in the basement membrane matrix [83,85,86]. It is worth mentioning here that some organoids have also been developed in suspension cultures in the absence of a reconstituted basement membrane [31,87,88]. These 3D structures have also been referred to spheroids due to the absence of a cell–substrate interaction. However, these structures could be referred to as organoids, as long as the aforementioned requirements are fulfilled [40].

PSC-derived organoids are used for modeling development, whereas ASC-derived organoids are crucial for understanding the process of tissue repair and maintenance in adult tissues [82]. In this regard, PSC- and ASC-derived organoids are complementary systems that represent the developmental and adult tissue stages, respectively. However adult stem cells can only be isolated from certain organs with a high regenerative capacity, such as intestinal epithelia that form organoids resembling polarized cysts; these organoids are not as complex as the organoids derived from PSCs. Nevertheless, the relative ease in establishing ASC-derived organoids makes them extremely useful for basic and clinical research (Table 3).

## 4. Liver Development and Regeneration: Cues from Nature

The human liver mainly consists of epithelial cells (hepatocytes and cholangiocytes) that work synergistically with stromal, endothelial, and mesenchymal cells to perform metabolic, exocrine, and endocrine functions crucial for homeostasis. In spite of fundamental biological differences, our understanding of human liver development and in vitro differentiation of liver progenitor/stem cells come from studies in mice. The posterior foregut endoderm gives rise to embryonic liver progenitor cells (hepatoblasts) during organogenesis. Furthermore, the surrounding mesenchyme secretes growth factors such as FGF, HGF, Wnt, and BMP, which facilitates hepatoblasts to proliferate and migrate to the adjacent mesoderm-forming liver bud [89]. Moreover, hepatoblasts also specify into hepatocytes and cholangiocytes [90] as Lgr5^+^ hepatoblasts are bipotent in nature [22]. The lineage commitment is influenced by local signals: hepatocytes adjacent to the portal mesenchyme give rise to cholangiocytes, whereas those further away are influenced by the signals from hematopoietic cells giving rise to hepatocytes. Furthermore, the adult liver remains under homeostatic condition to maintain normal physiological functioning. However, unlike the intestine with 3–5 days of cellular turnover, it takes 60 days for cholangiocytes and 150 days for hepatocytes to self-renew [91]. Moreover, homeostatic conditions are normally maintained via the division of mature cells [92,93]. The liver has a remarkable regenerative capacity, although repeated assaults could lead to fibrosis [94]. Nevertheless, when the liver is surgically resected, the remaining healthy tissue induces liver regeneration via tumor necrosis factor-alpha (TNFα) and interleukin-6 [95,96]. Hepatocytes lose their ability to undergo cell division and proliferation during toxin-mediated damage, such as viruses, alcohol-related fatty liver disease (ALD), and non-alcoholic fatty liver disease. However, the liver still maintains regenerative capabilities by activating ductal cells to repopulate the liver [97,98,99,100]. Clearly, these studies have substantially increased our understanding of liver biology, thus facilitating the establishment of in vitro models using stem/liver progenitor cells (Figure 2).

## 5. Liver Organoid Models

Various human cell and tissue models have been developed such as immortalized cell lines, xenografts, and organoids derived from differentiated stem cells to overcome the limitations posed by animal models. Every model has advantages and disadvantages; for instance, cells cultured in a conventional 2D system, where cells grow in a monolayer, are easy to maintain and respond well to genetic engineering tools, making them suitable for drug discovery and understanding disease progression. Nevertheless, 2D models lack cell–cell and cell–niche interactions that are crucial to the functioning of human organs in vivo [101]. These limitations could be addressed by using patient-derived xenografts that essentially recapitulate the heterogeneity and intricacy of the tissue of origin. But, xenografts are expensive to establish and are not suitable for high-throughput screening experiments [102]. This has led to a great deal of interest in organoids, which have been in use for almost two decades largely in the field of regenerative medicine [103,104].

In one of the earliest studies of liver organoids, cells isolated from newborn rats aggregated together and formed an organoid on a low-attachment plate. The organoid consisted of hepatocytes, forming central lumen-like structures, and cholangiocytes, forming bile ducts like structures. Interestingly the organoid also managed to increase in diameter over a period of time [5].

Progenitor cells split up from the foregut endoderm and from the liver bud during hepatogenesis, which is subsequently vascularized. To recapitulate the spatial arrangement of cells during organogenesis such as hepatogenesis, Taniguchi and colleagues co-cultured human-induced pluripotent stem cells specified into hepatocyte-like cells with mesenchymal and endothelial cells to form a liver bud. The human iPSCs after differentiation into hepatic endodermal cells using activin A followed by bFGF/BMP4 formed organoids after plating with endothelial cells and mesenchymal stem cells on Matrigel [42] (Figure 2). These organoids also developed blood vessels that connected to the host vessels within 48 h when transplanted into mouse models. Moreover, they were able to perform human-specific liver function and rescued mice from drug-induced liver failure [105].

### 5.1. Adult Stem Cell-Derived Liver Organoids

Ever since the establishment of crypt-villus organoids from LGR5^+^ stem cells [27], organoids have been developed from tissue-specific stem cells for various organs using a similar approach. For instance, hepatotoxin (carbon tetrachloride)-induced liver injury causes the Wnt-regulated LGR5^+^ stem cells to appear in the surroundings of bile ducts [30]. These cells give rise to cyst-like structures when cultured in the presence of Wnt agonist RSPO1. The cells could differentiate into hepatocytes in vitro and repopulate Fah^−/−^ mice upon transplantation [30]. Moreover, EpCAM^+^ biliary cell-derived human adult liver organoids were found to be genetically stable even after expansion for over a year in vitro [12]; consequently, these models could be used as indispensable tools in regenerative medicine, drug discovery, and understanding liver diseases. Nevertheless, the organoids are essentially epithelial stem cells that need to be differentiated into hepatocyte-like or cholangiocyte-like cells to be functional in vivo or in vitro [12].

### 5.2. Liver Organoids Derived from Primary Hepatocytes

The ability of hepatocytes to proliferate in vivo has been exploited to generate organoids that expand for months in vitro [34,106]. In one such study, organoids generated from mouse primary hepatocytes and human fetal hepatocytes formed ‘grape like’ structures normally seen in organoids developed from hepatocytes, as opposed to cyst-like structures formed by biliary epithelial organoids [34]. Interestingly, the organoids developed from hepatocytes exhibited regenerative characteristics after partial hepatectomy and had functional similarity to primary hepatocytes [34]. In another study, organoid models were developed to recapitulate liver cancer [107]. Intriguingly, inflammatory cytokines have been shown to expand organoids in vitro [35]. The expression profile of these organoids exhibited similarity to proliferating hepatocytes and repopulated livers of Fah^−/−^ mice [35]. Perhaps these protocols could be used to further optimize the culture conditions [34,35] for the expansion of organoids derived from an adult human liver. Moreover, hepatic organoids have also been derived from fibroblasts differentiated into hepatocytes by inducing the expression of hepatocyte-specific transcription factors [107]. However, PSC-derived hepatic organoids with increased generative capacity and patient specificity are advantageous over ASC-derived organoids for disease modelling and drug discovery [16].

### 5.3. hPSC-Derived Liver Buds

The ability of hPSCs to self-renew and differentiate into different types of cells has led to the development of numerous organoid models for liver and multi-organ organoids, where multiple cell types represent different human organs [40]. The methodology that was first used to develop liver organoids in vitro involved the generation of liver buds [42], where iPSC differentiated into hepatic endoderm cells were co-cultured with MSCs and human umbilical vein endothelial cells to emulate liver organogenesis [89]. Interestingly, the cells organized into a three dimensional liver bud on Matrigel had a gene expression profile similar to mouse embryonic liver buds [42]. The buds were able to vascularize and rescue a liver failure mouse model when engrafted [42]. Later, the same protocol was further improved and liver organoids were developed entirely from iPSC-derived endoderm, mesenchymal, and endothelial progenitor cells [108]. The organoids could be used as a potential source of cells for patients in regenerative medicine using iPSCs from HLA-matched donors and a differentiation medium that is of clinical grade. Nonetheless, the complexity of the co-culture system and a lack of maturity are limiting the clinical application of liver buds [42,108].

### 5.4. Cholangiocyte Organoids

Numerous studies have utilized signaling cues to direct the differentiation of hPSCs into cholangiocyte organoids, hepatic organoids, or hepatobiliary organoids having multiple types of cells [109]. The developmental signaling cues have been used to guide the differentiation of hPSCs into definitive endoderm followed by differentiation into definitive endoderm and ultimately into cholangiocytes with the induction of the notch signaling pathway [110]. The cholangiocytes were able to self-organize into organoids with structural similarities to the ductal structures found in the liver and expressed biliary marker genes [111,112,113] such as apical sodium-dependent bile acid transporter and cystic fibrosis transmembrane conductance regulator-mediated fluid secretion when embedded in Matrigel [112]. Interestingly, the organoids derived from human primary cholangiocytes retain plasticity and resume the expression of the transcriptional signature genes found in mature cholangiocytes when injected in the intrahepatic ducts of a liver undergoing ex vivo normothermic perfusion (EVNP), thus regenerating the functional biliary tree [114]. This proof-of-concept study opens new avenues in the field of regenerative medicine and disease modeling.

### 5.5. hPSC-Derived Hepatobiliary Organoids

Multi-cellular hepatobiliary organoids that consist of hepatocytes and cholangiocytes exhibiting bile duct-like morphology have also been developed by several groups [19,115,116], recapitulating the hepatic developmental process of an embryo. These organoids exhibiting the expression of hepatogenic genes and hepatobiliary functions were used to identify the role of JAG1 mutations in the development of liver disease [19]. In another study, iPSC-derived EpCAM^+^ endodermal cells were used to generate hepatic organoids that were able to expand for over a year and were used to characterize the role of the argininosuccinate synthetase (ASS1) enzyme in citrullinemia type 1 disease [117]. Hepatobiliary organoids have also been generated in a novel 3D suspension culture where the organoids self-organize and form functional bile canaliculi [116] and are thus useful models for studying cholestasis induced by drug or disease. The fact that these organoids are generated in a suspension culture in the absence of a matrix broadens their scope in the realm of cell therapy and drug screening [116].

### 5.6. hPSC-Derived Complex Liver Organoids

In a study aimed at recapitulating the complexity of the human liver by modeling steatohepatitis and fibrosis, pluripotent stem cell-derived organoids consisting of hepatocyte-like cells, Kupffer-like cells, and stellate-like cells were generated [18]. The organoids were able characterize steatohepatitis and fibrosis when treated with free fatty acids (FFA) for 5 days [18]. The protocol was further optimized to generate organoids with distinct bile canalicular lumen-like structures [118]. High-throughput screening was performed to assess the effects of drugs on liver injury [118]. The fact that these organoids were generated from PSCs differentiated into foregut progenitor cells which can be cryopreserved, and the induction of multiple cell types that took place simultaneously under the same culture conditions are the main advantages of this protocol. Further characterization using high-throughput screening can help us to better understand the developmental process of the human embryonic liver.

Hepato-biliary-pancreatic (HBP) organoids generated by co-culturing SOX2^+^ anterior spheroids in close proximity with CDX2^+^ posterior spheroids from hPSCs in the absence of signal cues were used to understand multi-organ communications during organogenesis [119]. The HBP organoids developed into multi-organ anlages consisting of hepatic and pancreatic tissues with connecting ducts [119]. This model provides an efficient and accessible way to understand the complex process of endoderm organogenesis in vitro. Models like these in combination with technologies such as organ-on-chip and tissue engineering could be developed to recapitulate organ-to-organ interaction and communication during the developmental process and disease onset [120,121,122].

## 6. Liver Organoid Validation

Both human and mice tissue samples have been used to establish cholangiocyte [12,30] and hepatocyte [34] organoids. Liver organoids are assessed based on measuring passaging time and proliferating organoid cells. The expression of hepatocyte-related markers (for example *ALB*, *HNF4A*, and *MRP4*) and cholangiocyte-related markers (such as *KRT19*, *KRT7*, and *SOX9*) are assessed using real-time quantitative PCR, and scRNA-seq on the RNA level and immunofluorescence staining is performed on the protein level. Liver progenitor makers (including LGR5) are often used to evaluate the differentiation state of organoids and *CYP3A4* and *CYP3A11* are used to assess organoid functionality on the RNA level. Moreover, low-density lipoprotein uptake, albumin secretion, assessment of glycogen accumulation with acid–Schiff staining, bile acid production, and CYP3A4 activity are observed to corroborate hepatocyte-specific functionality. Hepatocyte and cholangiocyte morphology is also examined using a transmission electron microscope (TEM). Ultimately, the integration and functionality of organoids observed in FRG immune-deficient mice potentially determines the functionality of generated organoids [30,35].

## 7. Validation of Tumor Organoids

It is imperative for the patient tissue-derived tumor organoids to maintain genomic, functional, and physiological profiles of the tissue of origin when compared. Comparing histological and immunohistochemistry profiles [123] of tumor-derived organoids with the tissue of origin as well as validation using genomic and transcriptomic profiles [124,125]. Tumor-derived organoids can be easily compared with origin tissue for cellular organization, protein expression, and tissue structure. Likewise, genomic and immunohistological analyses are used to validate tumor types and subtypes in PLC and other tumors [53,126,127]. Moreover, the confirmation of whether organoids have diverged at the genomic level is also important. In this regard, several studies conducted have shown that organoids retain mutations and copy number variations observed in parent tissue [124,125]. Nevertheless, sampling bias, spatial diversity, and intra-tumor heterogeneity could purportedly lead to greater genomic divergences [128,129]. Moreover, culture conditions might selectively favor certain clones, thus altering clonality over a period of time [127].

## 8. Application of Liver Organoids in Disease

The presence of hepatocytes, cholangiocytes, Kupffer cells, and endothelial cells makes liver a heterogenous organ [130]. Liver diseases are caused by the interplay between the environment and genetic factors, which is a manifestation of impaired communication between cell-cell and cell-environment [131,132,133]. Therefore, generating liver organoids with multiple cell types is imperative for modeling liver diseases, such as non-alcoholic fatty liver disease (NAFLD) and alcoholic liver disease (ALD).

### 8.1. Modeling Alcoholic Liver Disease

Consumption of high levels of alcohol has led to an increase in alcohol liver disease (ALD) and its related comorbidities globally. A wide range of liver pathologies including steatosis, fibrosis, cirrhosis, and even hepatocellular carcinoma are manifestations of ALD [134]. Unfortunately, due to limited access to liver tissue samples and a lack of reliable in vitro models until recently, there is no FDA-approved treatment for ADL [135]. However, a model system to recapitulate ALD in vitro was developed in 2019 by co-culturing hepatic organoids derived from hESCs with mesenchymal cells from a human fetal liver. In addition to the complex architecture displayed by these organoids, they exhibited functional similarity to mature hepatocytes by expressing enzymes involved in ethanol metabolism, inflammation, and fibrosis when treated with ethanol for a week [47]. ALD manifestations like oxidative stress and dysfunctional mitochondria were also observed in these organoids [47], making this system an efficient tool for drug discovery. Nevertheless, the addition of resident macrophages such as Kupffer cells would further improve the system.

### 8.2. Modeling Non-Alcoholic Fatty Liver Disease

The lack of approved pharmacological interventions to treat NAFLD (non-alcoholic fatty liver disease) and NASH (non-alcoholic steatohepatitis) could be attributed to the absence of proper animal and cellular models and limited access to liver biopsies from patients. To recapitulate the liver architecture and liver pathologies such as inflammation and fibrosis as observed in NAFLD, multicellular liver organoids consisting of stellate-like, hepatocyte-like, and Kupffer-like cells were developed and treated with FFA [18]. Numerous other models have been developed to model liver diseases (Table 2). 

Moreover, the treatment of organoids derived from a patient with Wolman disease, a genetic disorder which leads to steatohepatitis, with FGF19 rescued the cells [18]. Treatment with FFA induces a gene expression pattern similar to NASH in hepatobiliary organoids consisting of functional bile canaliculi. Furthermore, disruption of canaliculi system is observed in organoids recapitulating NASH [116]. The model was further improved when hiPSC-derived vascularized organoids consisting of hepatoblasts, MSCs, endothelial cells, and stellate cells were developed and exposed to FFA to mimic steatosis, inflammation, and fibrosis, as observed in NAFLD [136].

Generation of bipotent ductal organoids has also been achieved from liver biopsies of patients with NASH. These organoids exhibited increased lipid accumulation, lower levels of albumin, decreased ability to proliferate, and higher expression of cytochrome p450-related genes in comparison to the organoids derived from healthy livers [137]. The organoids thus generated mimicked hallmarks of NASH. Additionally, organoids from NASH mouse models with varying degrees of disease severity exhibited fibrosis and pro-inflammatory responses [138]. Patient-specific organoids generated from biopsies can potentially open new avenues in the field of precision medicine for liver diseases.

Interestingly, an individual’s predisposition to steatosis and its severity is dependent on the interplay between the environment and genetics [131,139]. Numerous genetic variations are associated with the progression of NAFLD [140], of which patatin-like phospholipase domain-containing protein 3 (PNPLA3) Ile148Met polymorphism is the most significant one [141]. Liraglutide, elafibranor, or momelotinib are shown to rescue spheroids consisting of immortalized hepatic stellate cells (LX-2) and HepG2, that are homozygous for the PNPLA3 Ile148Met variant, from fat and collagen accumulation when exposed to FFA [142,143]. A similar strategy was used to understand the association of NAFLD with transmembrane 6 superfamily member 2 (TM6SF2) Glu167Lys and membrane-bound *O*-acyltransferase 7 (MBOAT7) [144,145]. Patient-derived organoids can potentially be used to assess the effects of genetic polymorphism solely on NAFLD progression to NASH without the interference of environmental factors.

### 8.3. Organoids in Viral Infection

Microbial diseases are taking a toll on human health globally. The health emergency caused by Zika virus in addition to the recent COVID-19 pandemic are stark reminders of how infectious agents pose a threat to human health worldwide. Organoids can be instrumental in understanding infectious diseases where research on animal models is hindered by species-specific differences. Since wild-type mice are not susceptible to SARS-CoV-2, the COVID-19 pandemic resulting in almost 7 million deaths has increased the prospects of using organoid models for disease modeling [70] (Figure 3). This is attributed to the fact that organoids are ideally suited to study the implications of viral infection on multiple organs [146]. Although COVID-19 caused by SARS-CoV-2 mostly affects the lungs and upper respiratory tract, it has also been found to affect the liver, blood vessels, heart, intestine, kidneys, and even the brain. A number of organoid and organ-on-chip models have been developed to mimic the human respiratory tract for studying COVID-19 and tissue response [146] (Table 2). While the angiotensin-converting enzyme 2 (ACE2) receptor for SARS-CoV-2 is ubiquitous in the respiratory tract, the expression is varied with a higher expression found in the nasal epithelial of the upper respiratory tract compared to bronchioles and alveoli [147]. Organoids consisting of cholangiocyte progenitors expressing ACE2 and transmembrane serine protease 2 (TMPRSS2), an activator of the viral spike protein, are shown to be susceptible to SARS-CoV-2 infection [13]. A section of patients have exhibited multi-organ damage [148,149,150]; notably, liver damage has emerged as a co-existing symptom, as observed in a recent epidemiological study carried out in Shanghai (China), where 75 of 148 COVID-19 patients had abnormal liver functioning evident by higher amounts of alanine aminotransferase (ALT), aspartate aminotransferase (AST), alkaline phosphatase (ALP), and total bilirubin (TBIL) [151]. The fact that a significant proportion of cholangiocytes have been found to be ACE2+ in healthy livers [152] renders them susceptible to SARS-CoV-2. In a recent study by Zhao et al., human liver ductal organoids were found to be susceptible to SARS-COV-2. Moreover, the organoids also supported viral replication [13]. Consequently, the infection induced apoptosis in cholangiocytes via the CD40 molecule (CD40), caspase recruitment domain family member 8 (CARD8), and serine/threonine kinase 4 (STK4). Moreover, performing gene set enrichment analysis (GSEA) showed disruption in cell junction enrichment genes, thus interfering with the role of ductal epithelia as a permeability barrier [13]. Interestingly, SARS-CoV-2 also significantly decreases the expression of solute carrier family 10 member 2 (SLC10A2) and the cystic fibrosis transmembrane conductance regulator (CFTR), which are involved in bile acid transport. This implies the damage to cholangiocytes as a major factor in liver injury in COVID-19 patients [13].

Mild-to-moderate steatosis and lobular/portal activity has also been observed in liver biopsy specimens obtained from patients deceased due to severe COVID-19 infection, suggesting COVID-19 or drugs as a source of liver injury [153]. The injury could be attributed to a number of reasons. Inflammation due to COVID-19 could potentially trigger the immune system causing liver damage [154], as higher levels of the C reactive protein (CRP), serum ferritin, LDH (lactate dehydrogenase), D-dimer, IL-2, and IL-6 are found in patients with severe COVID-19 infection [155]. The presence of ACE2^+^ cells in liver makes it a potential target, thus leading to viral-associated cellular cytotoxicity [152]. However, injury due to direct viral replication has yet to be observed. Anoxia-induced hypoxic hepatitis has been frequently observed in severe cases. Finally, drug-induced liver injury has also been observed in patients. The lack of knowledge about the effectiveness of anti-viral drugs being used against COVID-19 has led to the administration of numerous anti-viral drugs such as lopinavir/ritonavir, remdesivir, chloroquine, tocilizumab, uminefovir, and Traditional Chinese medications. This has led to an increase in drug-related liver injuries [156]. SARS-CoV-2 may also exacerbate pre-existing chronic liver diseases, making patients with these conditions more prone to liver damage [157]. Nevertheless, severe liver injuries are rare [156]. The implication of non-alcoholic fatty liver disease (NAFLD) in patients has also been observed in a recent study [158]. Over 50–75% of patients with COVID-19 and NAFLD have some form of liver injury, although most of the injuries were mild patients with NAFLD who are more prone to progress to the severe form of COVID-19 infection and take longer to recover [158].

Organoids have also been used for testing antiviral drugs for hepatic viruses with narrow tropism. Organoids derived from fetal and adult liver have been successfully used for the replication of hepatitis E virus (HEV), a single-stranded RNA virus. Liver-derived organoids fully supported the life cycle of HEV virus and the apical secretion of HEV particles was observed when the organoids in the transwell system were directed toward polarized monolayers. Robust host responses were triggered by the viral replication when genome-wide transcriptomic analyses were performed. The successful recapitulation of HEV infection in host-derived organoids and drug screening identified brequinar and homoharringtonine as potent inhibitors of even a ribavirin-resistant HEV variant with G1634R mutation [49]. Additionally, in vitro host virus interaction was also observed in iPSC-derived liver organoids and hepatitis B virus (HBV), leading to the downregulation of hepatic genes such as HNF1A, ALB, G6PC, CYP3A7, CYP2C9, and RBP4 in organoids infected with HBV [50]. Moreover, EpCAM^+^ stem cells isolated from liver resections used for the generation of liver organoids have been successfully co-cultured with patient-derived CD8+ T cells using a microfluidic technique to recapitulate the patient-specific immunological response against HCV. CD8+ T cells successfully killed liver organoids that were pulsed with HCV non-structural protein 3 (NS3)-specific peptide [52].

### 8.4. Tumor-Derived Organoids

A wide variation has been observed in the past decade in organoid derivation, including preferred tumor source and the downstream processes. Primary tumors [159], metastasized tumors [160], circulating tumor cells [161], and tumor cells isolated from liquid effusions [162], including liquid and solid biopsies, autopsies, and surgical resections [163], have all been used to generate tumor organoids. Samples once collected are processed before encapsulation in the 3D matrix for the downstream cultures (Figure 2 and Figure 3). The two most prevalent strategies for pre-processing the tissue samples currently used are (1) the dissociation of tissue samples into single cells followed by encapsulation and (2) mechanical and enzymatic mincing of tissue samples into smaller fragments and encapsulation of tumor fragments that are a few millimeters in size. An alternate method used to generate cancer organoids is to induce driver mutations that recapitulate cancer onset and progression. Although each strategy for the derivation of the organoid culture potentially helps in demystifying certain unknown aspects of cancer biology, a lack of standardization puts their use in clinical research under jeopardy [20]. Liver cancer lacks dependable in vitro models to recapitulate cancer pathophysiology. Culture systems as mentioned above have been used for the long-term expansion of primary liver cancer (PLC) organoids derived from common PLC subtypes: cholangiocarcinoma (CC), hepatocellular carcinoma (HCC), and combined hepatocellular–cholangiocarcinoma (CHC) tumors (Table 2). These PLC-derived organoids maintain the genetic landscape and histological architecture of the parent tumors even after a long-term culture. This has also been observed in xenograft studies for PLC-derived organoids. The organoids possessed tumor-specific biomarkers and were instrumental in the identification of ERK inhibitor SCH772984 as a therapeutic agent for liver cancer [53].

## 9. Organ-on-Chip

The fact that the data obtained from animal models regularly fail to predict the outcome of human trials and the lack of preclinical models that are human-relevant call for finding alternatives to animal models [164,165,166,167,168]. This has also led to fewer effective drugs and an unsustainable rise in healthcare [169]. Organ-on-chip, a microfluidic culture device that recapitulates organ physiology and function in vitro, represents an alternative to the animal model. These microphysiological systems come in various shapes and sizes that consist of hollow channels lined by organ-specific cells under dynamic fluid flow (Figure 4). Multi-organ systems, such as body-on-chip, can also be created by coupling two or more chips fluidically to mimic the physiology of the human body. Advancement in stem cell technology and patient-derived organoids has opened new possibilities in the field of personalized medicine. As the design and functionality of microfluidic devices improve, the next step would be to prove the advantage of microfluidic systems over animal models [169].

Numerous liver chip designs to model hepatotoxicity, inflammation, drug–drug interactions, and infection have been developed in this regard. Organ-on-chip consisting of multiwells lined by hepatocytes and Kupffer cells mimicked the glucocorticoid hydrocortisone breakdown to phase I and phase II metabolites in correspondence with the human data [170]. In another similar system based on microfluidic liver chip IL-6-induced inflammation was shown to suppress cytochrome P4503A4 isoform (CYP3A4) activity, increase secretion of the C-reactive protein, and reduce IL-6 receptor shedding [171]. Treatment with tocilizumab (anti-IL-6 receptor monoclonal antibody) modulated the activity of CYP3A4, therefore altering the metabolism of simvastatin hydroxy acid and thus emulating the human drug–drug interaction which is otherwise not observable in 2D models. Another interesting study explored the effects of human population variability on hepatic drug metabolism using multiple liver chips each lined by hepatocytes from a different donor [172]. Analysis of metabolic depletion profiles of six drugs confirmed the existence of substantial inter-donor variability with respect to the gene expression levels, drug metabolism, and other hepatocyte functions. Importantly, clearance values predicted on-chip correlated well with those observed in vivo, and a physiologically based pharmacokinetics model developed for lidocaine successfully predicted the observed clinical concentration–time profiles and associated population variability. The liver chip has also been used for modeling viral infection. Hepatitis B virus infection of the hepatocyte-lined liver chip successfully modelled the viral life cycle including the replication and maintenance of covalently closed circular DNA (cccDNA) [51]. Additionally, the model recapitulated the host cytokine and innate immune response observed in patients infected with HBV [169].

## 10. Organoids in Precision Medicine

Human tumors can be used for the generation of organoids as an individualized precision medicine approach in cancer treatment [173] (Figure 4). The advancement in the field of 3D culture has made it possible to generate organoids for many epithelial-adult and pediatric tumors. Patient cancer-derived organoids successfully retain the genetic, histopathological, and drug response of original tumors and can be generated with a higher success rate compared to 2D cultures [173,174]. This allows for individualized targeted therapies [175]. Nevertheless, limitations like higher costs and varying success rate need to be overcome for cancer organoids to transition from the bench to the bedside [173,174]. Additionally, a lack of tumor microenvironment that includes endothelial cells, fibroblasts, stromal, and immune cells potentially limits their use in predicting patient responses. On the other hand, species-specific differences in immune response limits the use of xenografts due to their reliance on the immune system of the mouse. Therefore, complementing tumor organoids by immune cells could potentially overcome the shortcomings of organoid technology and help better understand the role of the immune system in human cancer [176].
Figure 4Organoids in translational medicine. (**a**) Drug-Screening; Patient-derived organoids can be used in a high-throughput drug screening platform to study DILI. These pre-clinical platforms can be used as diagnostic tools and predictive models for drug discovery [177]. (**b**) Organ-on-chip; Depiction of a mechanically actuable two channel organ chip fabricated using soft lithography from PDMS with two channels parallel to each other separated by a microporous membrane. Cells from different tissues are cultured on top and bottom of the extracellular matrix (ECM)-coated microporous membrane to emulate tissue–tissue interaction where air introduction above the epithelium can take place to recreate air-liquid interface (as observed in lungs) or the channel could be used for fluid perfusion [169]. Human liver organoid-based chips have recently been used for DILI risk assessment [177]. DILI, drug induced liver injury; PDMS, polydimethylsiloxane.
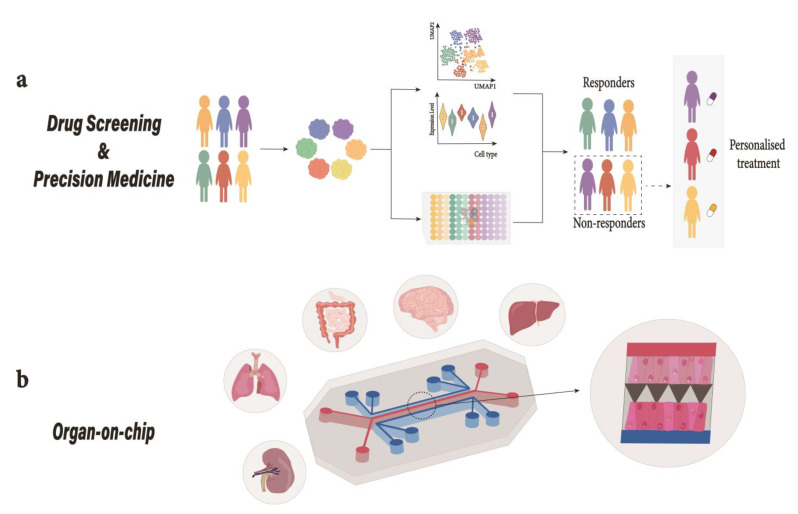


There are two main approaches to co-culturing immune cells with cancer organoids. One is a holistic approach where endogenous immune cells present in the tumor are preserved and truly depict a patient immune response to a particular tumor; alternatively, the other is a reductionist approach that involves patient tumor organoids co-cultured with immune cells propagated separately [176,178], allowing for better understanding of tumor–immune interaction even though the immune cell repertoire is limited. One such example includes the co-culturing of peripheral blood-derived T cells with colorectal and non-small cell lung cancer organoids [179]. A potential use of these complex models is to study immunotherapy regimens, a new class of therapy that utilizes patient’s immune system to fight tumor cells [178]. Organoids generated from patient tumor have been co-cultured with genetically engineered chimeric antigen receptor (CAR) T cells expressing antigen receptors specific to the antigens present on cancer cells [180]. CAR-T therapy has so far been successful against hematological malignancies; however, treatment of solid tumors, including liver cancer, remains a challenge due to a lack of specific antigenic targets, tumor heterogeneity, and the immunosuppressive tumor microenvironment [181,182]. Organoid technology could also be harnessed to produce immune cells for immunotherapy in addition to modeling different cancer types [183]. Organoid-derived cells can integrate, and under certain circumstances, restore organ function [184]. Disease treatment using organoid-derived cells could potentially be a reality in the near future; in fact, exciting new opportunities await us with the beginning of the first clinical trial for head and neck cancer treatment using organoid-derived salivary gland cells (ClinicalTrials.gov Identifier: NCT04593589).

An approach to predicting the efficacy of individualized treatments is to use patient-derived organoids (Table 3). For instance, organoids derived from patients with cystic fibrosis (CF) have been used to predict the effectiveness of the treatment [81,82,185]. Owing to the self-renewable characteristics of organoids, they could potentially serve as cryopreserved biobanks of both healthy and diseased individuals. This could be made possible with the establishment of biobanks, such as HUB organoids by Hubrecht Institute. Such biobanks can help with the development of personalized treatments, drug discovery, and identifying new diagnostic markers for liver diseases [159,186]. As a matter of fact, NAFLD progression differs among humans [131,139], and so individuals at risk of developing NAFLD could be identified by generating organoid models with genetic variations from different sources (Figure 3). Moreover, the organoids could also be used for developing precision medicine and predicting drug-induced genetic and nongenetic risk factors that lead to acute liver failure [187,188].

## 11. Gene Therapy and Regenerative Medicine

Gene editing tools such as CRISPR-Cas9 used in cell/animal models could potentially prevent or rollback phenotypes associated with genetic diseases [189]. Modern genome editing techniques in conjunction with stem cell technology are being used to generate diseased organoid models from isogenic cell lines as well as organoids to treat metabolic diseases due to genetic mutations [190,191]. One of the key advantages of generating patient-specific organoids is to overcome allograft rejection. The expression of UCP1 in human white preadipocytes using CRISPR-Cas9 led to the generation of brown-like adipocyte cells that improved glucose tolerance and insulin sensitivity in obese mice upon transplantation [192]. Likewise, the deletion of NRIP1, a thermogenic suppressor gene, using CRISPR-Cas9 and subsequent implantation into obese mice enhanced glucose tolerance and decreased liver triglyceride levels in addition to decreasing adiposity [193]. Interestingly, Alagille syndrome and Wilson disease-related mutations have been successfully reverted to the wild type using CRISPR-Cas9 base editing technology in patient-derived organoids [19,194], thus opening new avenues in approaching liver-related metabolic disorders. Even though an orthotopic liver transplant is marred by the high risk of immune rejection and a lack of donors, it is still the only viable option for late-stage liver failure [195]. A viable alternative approach is to generate liver organoids consisting of complex structures that are able to imitate liver function. In fact, liver organoids have shown promising results when engrafted into the mice liver [34,35,42,108].

## 12. Current Limitations of the Panacea Illusion

Even though organoid models have been instrumental in providing insights into human-centric aspects of development and disease and have the potential to radically revolutionize drug discovery and contribute toward clinical treatment, albeit with some caveats that need consideration. Hypoxia and a lack of access to nutrients result in necrotic cores in large organoids; this is mainly due to the absence of a functional vascular system in organoids [196]. Efforts are underway to increase the supply of oxygen and nutrients to the organoid core by introducing blood vessels [197]. Apart from being a source of transportation, blood vessels play key role in delivering signaling cues to facilitate cell migration and differentiation [198]. Varying degrees of success has been achieved in generating networks of blood vessels and endothelial cells by employing different strategies including organoid transplantation into mice models [197], thus leveraging endogenous cells or by co-culturing organoids with exogenous endothelial cells [198]. However, unless these co-cultures could be established in a robust and cost-effective way, it is essential to remove stressed cells to avoid bias from oxidative stress [199].

A disadvantage of organoid models is the requirement of exogenously supplemented animal-derived ECM which severely limits their clinical use and hinders transplantation into humans. Matrigel, which is purified from Engelbreth–Holm–Swarm (EHS) sarcoma, is the most widely used basement membrane that contains a considerable amount of ECM components, primarily collagen IV, laminin, and entactin. Additionally, it contains growth factors such as insulin-like growth factor 1 (IGF-1), EGF, TGF-b, and platelet-derived growth factor (PDGF). The presence of thousands of proteins, lot-to-lot variations in composition, and differences in the mechanical aspects of Matrigel preparations results in batch-to-batch divergences in organoid cultures [88]. Moreover, there are ethical considerations regarding Matrigel production which requires growing tumors in mice. This has led to efforts being made to replace Matrigel with synthetic hydrogels, decellularized ECM from organs, and isolated ECM proteins [88,200]. Decellularized ECM purportedly retains intrinsic signaling and mechanical cues; however, the availability of human organs limits their use [88]. Synthetic hydrogels or specific components of ECM are also used as more defined approaches [200]. Particularly, unprecedented flexibility is introduced in organoid cultures via synthetic hydrogels as multiple strategies could be used to modulate their chemical and mechanical attributes; for instance, the generation of localized complex gradients consisting of growth factors. Nevertheless, further technological development is required to utilize the full potential of synthetic hydrogels [88].

Altogether, organoids models have led to a great degree of understanding into the developmental processes and disease, especially the mechanisms of genetic diseases. They have been vital in demonstrating how human organs are affected by infectious diseases and have been instrumental in generating valuable insights during Zika virus and the COVID-19 pandemic. Organoids have more recently entered clinical setups. The first organoid-based personalized medicine regimen in the Netherlands consisted of rectal organoids from CF patients to predict their response to various drugs [185]. Such approaches would more likely be used for other diseases including cancer in the near future. Similarly, transplantation of salivary gland-derived cells (ClinicalTrials.gov Identifier: NCT04593589) will open new opportunities in the field of cell replacement therapy. The future looks bright for the field of transplantation and regenerative medicine. Transplantation of neuronal progenitor cells, skin grafts for burns, and endless other possibilities could revolutionize the field of regenerative medicine.

## 13. Conclusions

A lack of in vitro models that reliably recapitulate human liver physiology has been a limiting factor in the transition from basic to translational research. Organoids derived from hPSCs and/or patient biopsies are excellent platforms to study disease development and progression and provide technological advancement in the realm of personalized medicine. Further advancements are required in current organoid technology by improving cellular composition to better emulate liver heterogeneity and function. The insights gained from organoids can further our understanding of liver development and have the potential to improve personalized therapy and drug development.

## Figures and Tables

**Figure 1 ijms-24-15921-f001:**
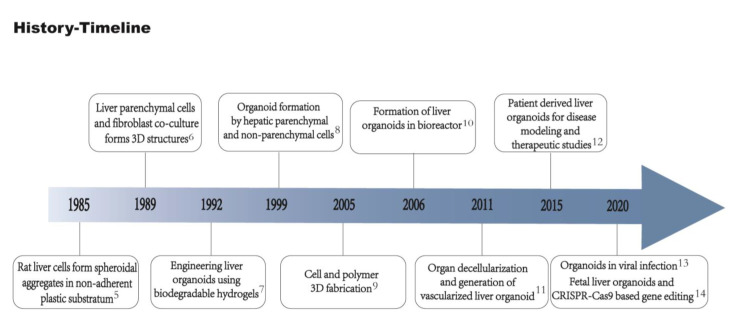
Timeline of important events in liver organoid technology. Rat liver cells successfully formed spheroidal aggregates in non-adherent plastic substratum [5]. Hepatocytes and fibroblasts co-culture formed 3D structures [6]. Biodegradable hydrogels used to generate liver organoids in vitro [7]. Liver parenchymal and non-parenchymal cells isolated and co-cultured to form organoids [8]. 3D Cell and biomaterial complex formation [9]. Development of bioreactor for the generation of liver organoids using bio-artificial liver [10]. Generation of vascularized liver organoids [11]. Generation of organoids from patient-derived cells [12]. Generation of liver ductal organoids to recapitulate SAR-CoV-2 infection [13]. Development of fetal liver organoids for gene editing using CRISPR-Cas9 [14].

**Figure 2 ijms-24-15921-f002:**
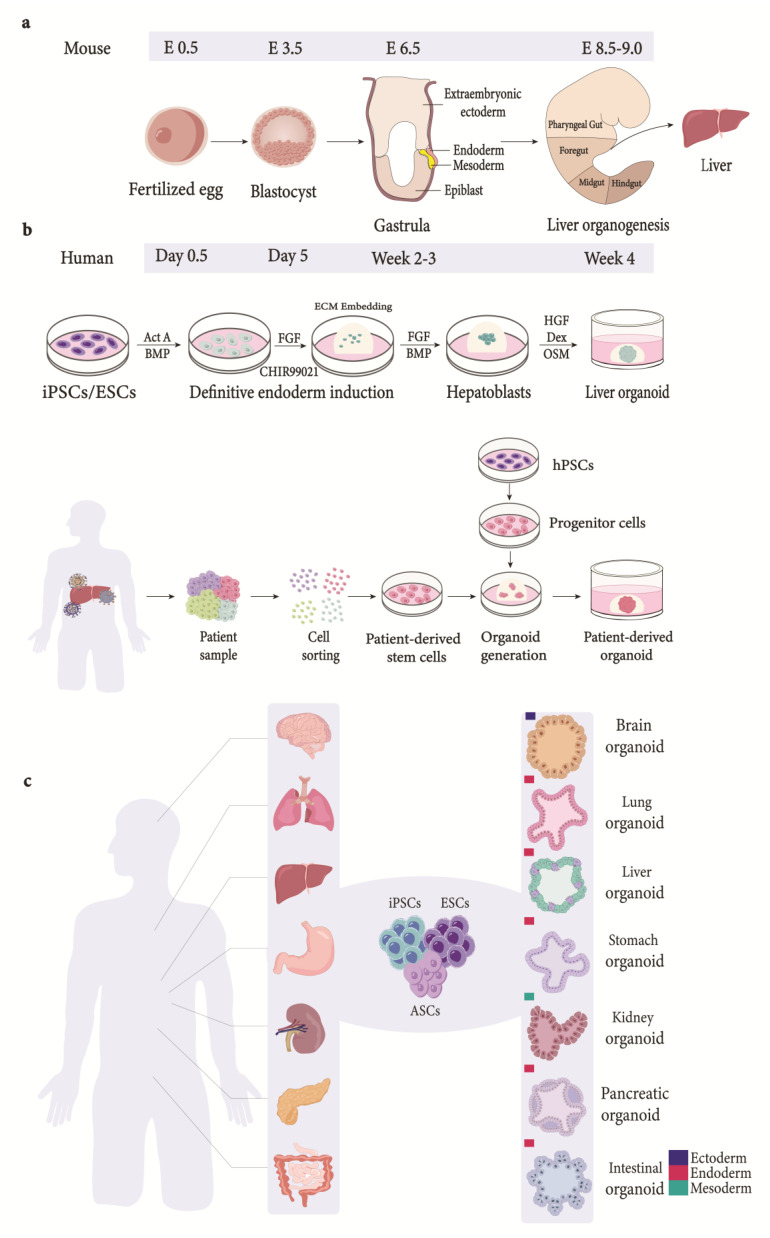
Organogenesis and generation of liver organoids from various sources. (**a**) Schematic depiction of stages involved in organogenesis. Embryo formation is followed by blastocyst stage. The blastocyst consists of an outer layer of trophectodermal cells and ICM. The ICM, consisting of embryonic stem cells, further specializes into either epiblast lineage or primitive endoderm lineage during the late stages of its formation. Blastocyst is followed by gastrulation, where morphological rearrangements transform epiblast into the three germ layers: ectoderm, endoderm, and mesoderm [16,17]. The endoderm becomes patterned into anterior foregut (AF), posterior foregut (PF), midgut (M), and hindgut (H). As illustrated above, the liver is derived from PF domain of the endoderm. (**b**) Organoids derived from pluripotent stem cells follow a stage-wise differentiation process that recapitulates signaling pathways observed during development. The differentiation process starts by directing iPSCs/ESCs towards endodermal fate when exposed to Act A and Wnt. The cells are embedded in ECM and differentiated into hepatoblasts-like cells (progenitor cells) using FGF and BMP. Hepatoblast-like cells differentiate into hepatocyte-like cells via exposure to OSM [18,19]. Moreover, ductal organoids can be generated by modulating FGF, EGF, and Act A signaling [16]. Hepatoblasts embedded in ECM give rise to hepatic organoids. Several tissue sources have been used to generate patient-derived organoids using a number of different techniques for tissue processing. The variation in techniques used has led to non-standardized patient-derived organoid culture techniques [20]. In general, tissue-derived stem cells are dissociated into single cells and embedded in extracellular matrix to generate organoids [21,22]. (**c**) Generation of organoids. Organoids generated for various organs derived from the three germ layers. ICM, inner cell mass; Act A, Activin A; BMP, bone morphogenetic protein; FGF, fibroblast growth factor; ECM, extracellular matrix; HGF, hepatocyte growth factor; OSM, Oncostatin M; Dex, dexamethasone; iPSCs, induced pluripotent stem cells; ESCs, embryonic stem cells; hPSCs, human pluripotent stem cells; ASCs, adult stem cells.

**Figure 3 ijms-24-15921-f003:**
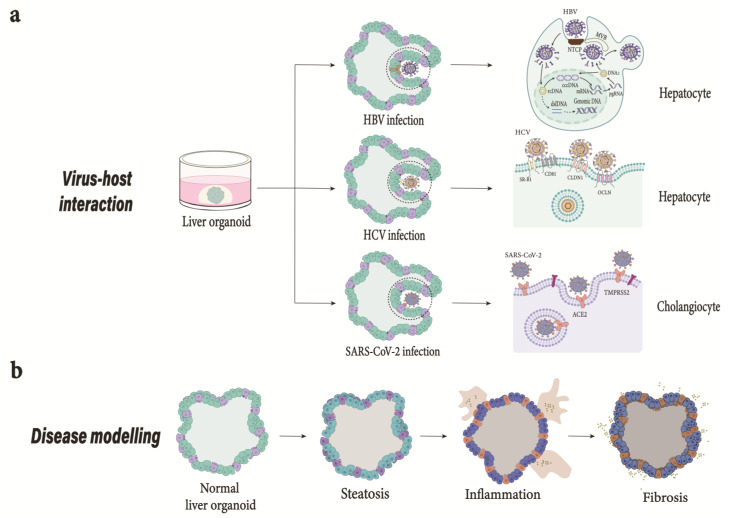
Application of liver organoid models. (**a**) Virus host interaction. HBV; Modeling HBV infection and drug screening. The model developed can successfully recapitulate HBV infection [21]. Depiction of HBV life cycle from attachment to secretion of viral particles. HBV enters hepatocyte via a mechanism involving NTCP receptor. Once internalized, the HBV’s rcDNA genome liberated into the nucleus is converted into cccDNA, which serves as a template for transcription. The dslDNA produced can either integrate into cellular genome or convert into cccDNA. The viral mRNA transported to cytoplasm is translated into viral proteins. The pgRNA and viral polymerase are encapsulated and reverse transcribed into progeny rcDNA within the nucleocapsid. HCV; The importance of new models to study HCV is crucial due to limited animal models [52]. Organoids derived from adult stem cells and hPSCs can be used to further our understanding of HCV infection and develop antiviral drugs. HCV interaction with cell surface receptors initiates viral entry. HEV; Generation of liver-derived organoids support HEV infection and life cycle and could help develop new therapies. SARS-CoV-2; The interaction of spike protein with ACE2 receptor in the presence of TMPRSS2 facilitates the entry of SARS-CoV-2 into the host cell. COVID-19 caused by SARS-CoV-2 could mediate cell damage, dysregulate RAAS leading to decreased cleavage of angiotensin I and angiotensin II, thromboinflammation, and endothelial cell damage, and inhibit interferon signaling, i.e., depletion of T lymphocytes and production of cytokines such as IL-6 and TNFα [76]. (**b**) Modeling steatohepatitis in vitro. The failure of animal models in identifying translatable therapies highlights the need for improved models. Generation of organoids from patient-derived hPSCs consisting of multiple hepatic cell types successfully emulates liver-in-a-dish and can be utilized to study liver inflammation and fibrosis and identify effective drug treatments [18]. NTCP, Sodium taurocholate cotransporting polypeptide; rcDNA, relaxed circular DNA; cccDNA, closed circular DNA; dslDNA, double-stranded linear DNA; pgRNA, pregenomic RNA; ACE2, Angiotensin-converting enzyme 2; TMPRSS2, transmembrane protease, serine 2; RAAS, renin-angiotensin-aldosterone system; IL-6, interleukin 6; TNFα, tumor necrosis factor alpha.

**Table 3 ijms-24-15921-t003:** Organoids in clinical trials.

Organ	NCT Number	Study Title	Study Status	Conditions	Study Objective
** Liver **	NCT05183425	Patient-derived Organoids Predicts the Clinical Efficiency of Colorectal Liver Metastasis	Recruiting	Explore the Consistency of Drug Sensitivity Between Primary Colorectal Cancer and Liver Metastases	Evaluate consistency of drug sensitivity of primary tumor and its matched liver metastasis
** Pancreas **	NCT04777604	Development of a Prediction Platform for Neoadjuvant Treatment and Prognosis in Pancreatic Cancer Using Organoid	Not yet recruiting	Pancreas Cancer	Survival rate of pancreatic cancer patients after they were started adjuvant chemotherapy
	NCT04736043	Development of a Prediction Platform for Adjuvant Treatment and Prognosis in Resected Pancreatic Cancer Using Organoid	Recruiting	Pancreatic Cancer Resectable	Overall survival rate after adjuvant chemotherapy for pancreatic cancer was started
	NCT05351983	Patient-derived Organoids Drug Screen in Pancreatic Cancer	Recruiting	Pancreas Cancer |Pancreas Neoplasm|Pancreas Adenocarcinoma|Pancreatic Cancer|Pancreatic Neoplasms|Pancreatic Adenocarcinoma	Feasibility of patient-derived organoids that can be successfully used for drug screening
	NCT04931381	Organoid-Guided Chemotherapy for Advanced Pancreatic Cancer	Recruiting	Advanced Pancreatic Cancer	Measure patient response to cancer treatment, complete response (CR), partial response (PR), or stable disease (SD)
	NCT04931394	Organoid-Guided Adjuvant Chemotherapy for Pancreatic Cancer	Recruiting	Pancreatic Cancer	Radiology assessment of response to adjuvant therapy
	NCT05842187	In Vitro Organoid Drug Sensitivity-Guided Treatment for Metastatic Pancreatic and Gastric Cancer	Recruiting	Pancreatic Cancer|Gastric Cancer	Analyze PFS (Progressive-free survival) from initiation of treatment to the occurrence of disease progression or death
** Brain **	NCT05772741	Grafts of GSCs Into Brain Organoids for Testing Anti-invasion Drugs	Recruiting	Glioblastoma|Glioma, Malignant	Study anti-invasive drugs against invasive glioma stem cells co-cultured with brain organoids
** Kidney **	NCT04342286	To Establish a Reproducible Organoid Culture Model with Human Kidney Cancer	Completed	Kidney Cancer	Establishment of patient-derived kidney tumor organoids with phenotypic stability
** Intestine **	NCT04497727	Gut Organoid Study	Active, not recruiting	Gut Inflammation	Gut–microbiota interaction in hypertensive and normotensive reference subjects
	NCT03256266	Effect of Antigens or Therapeutic Agents on in Vitro Human Intestinal Organoids	Recruiting	Intestine Disease	Gluten peptide provocation to determine proliferation, apoptosis, histology, and cytokine expression
	NCT05294107	Intestinal Organoids	Recruiting	Digestive System Diseases |Inflammatory Bowel Disease, Ulcerative Colitis Type|Crohn Disease	Patient-derived organoids cultured on 3D matrix gel for molecular screening to assess the effects of intestinal stress and healing
	NCT02874365	Intestinal Stem Cells Characterization	Recruiting	Inflammatory Bowel Diseases	Generation of organoid models for stem cell characterization
	NCT05832398	Precision Chemotherapy Based on Organoid Drug Sensitivity for Colorectal Cancer	Recruiting	Colorectal Cancer	Organoid generation for drug screening
	NCT05425901	Preclinical Evaluation of Multimodal Therapeutic Strategies in Intestinal Irradiation and Inflammatory Bowel Disease from Organoids	Not yet recruiting	Radiation Enteritis|Inflammatory Bowel Diseases	Optimize organoid generation to mimic human intestine by performing scRNA-seq
	NCT04371198	Patient-Derived Organoids for Rectal Cancer	Completed	Rectum Cancer	Patient-derived organoid to study rectal cancer
	NCT04996355	Organoids-on-a-chip for Colorectal Cancer and in Vitro Screening of Chemotherapeutic Drugs	Recruiting	Colorecta l Neoplasms|Organoids	Chemotherapeutic drug screening based on organoid chips to predict the effect of chemotherapy against Colorectal cancer
	NCT05352165	The Clinical Efficacy of Drug Sensitive Neoadjuvant Chemotherapy Based on Organoid Versus Traditional Neoadjuvant Chemotherapy in Advanced Rectal Cancer	Not yet recruiting	Neoadjuvant Therapy	Neoadjuvant therapy in organoids
	NCT05183425	Patient-derived Organoids Predicts the Clinical Efficiency of Colorectal Liver Metastasis	Recruiting	Explore the Consistency of Drug Sensitivity Between Primary Colorectal Cancer and Liver Metastases	Organoids in testing drug sensitivity to primary tumor and its matched liver metastasis
	NCT05304741	The Culture of Advanced/Recurrent/Metastatic Colorectal Cancer Organoids and Drug Screening	Recruiting	Colorectal Cancer	Generate patient-derived organoids to evaluate overall survival
	NCT04906733	Cetuximab Sensitivity Correlation Between Patient-Derived Organoids and Clinical Response in Colon Cancer Patients.	Recruiting	Colon Cancer|Organoids|Metastatic Cancer	Association of drug sensitivity tests in patient-derived organoids and clinical outcomes
** Lungs **	NCT04859166	Prospective Primary Human Lung cancer Organoids to Predict Treatment Response	Completed	Lung Cancer	To successfully establish PDX model and biobanks of lung cancer organoids and determine the frequency of primary, secondary, and tertiary organoid formation
	NCT05669586	Organoids Predict Therapeutic Response in Patients with Multi-line Drug-resistant Non-small Cell Lung Cancer	Recruiting	Lung Cancer|Organoid|NSCLC	Generate non-small cell lung cancer organoids to predict therapeutic response in multi-drug resistant cancer and evaluate progression-free survival
	NCT03655015	Patient-derived Organoid Model and Circulating Tumor Cells for Treatment Response of Lung Cancer	Recruiting	Lung Neoplasm	Establish biobank of patient-derived organoids (PDOs) to recapitulate ex vivo responses observed clinically
	NCT05092009	Lung Cancer Organoids and Patient-Derived Tumor Xenografts	Recruiting	Lung Cancer	Generate normal and patient tumor-derived organoids for long-term culture and bio-banking
	NCT03979170	Patient-derived Organoids of Lung Cancer to Test Drug Response	Recruiting	Lung Cancer|Organoid	Establish patient-derived organoids that are phenotypically and genetically identical to the source tumor
	NCT05136014	Evaluation of the Response to Tyrosine Kinase Inhibitors in Localized Non-small Cell Lung Cancer (NSCLC) Patients with EGFR Mutation in a Patient-derived Organoid Model	Enrolling by invitation	Lung Cancer|Lung Adenocarcinoma|EGFR Activating Mutation| KRAS Mutation-Related Tumors|Non-Small Cell Lung Cancer	In vivo evaluation of osimertinib alone or in combination in patient-derived organoids
** Gastric **	NCT05842187	In Vitro Organoid Drug Sensitivity-Guided Treatment for Metastatic Pancreatic and Gastric Cancer	Recruiting	Pancreatic Cancer|Gastric Cancer	Pancreatic and gastric metastatic cancer organoids to study drug sensitivity and PFS (Progressive-free survival)
	NCT05652348	Response Prediction of Hyperthermic Intraperitoneal Chemotherapy in Gastro- Intestinal Cancer	Recruiting	Gastric Cancer|Colon Cancer|Peritoneal Carcinomatosis	Ex vivo response to dose dependent chemotherapeutic agents in organoid models
	NCT05351398	The Clinical Efficacy of Drug Sensitive Neoadjuvant Chemotherapy Based on Organoid Versus Traditional Neoadjuvant Chemotherapy in Advanced Gastric Cancer	Not yet recruiting	Advanced Gastric Carcinoma	Determine clinical efficacy based on studies in advanced gastric cancer organoids by evaluating Objective Response Rate (ORR)
	NCT05203549	Consistency Between Treatment Responses in PDO Models and Clinical Outcomes in Gastric Cancer	Recruiting	Gastric Cancer	Establish and optimize generation of patient-derived organoids from gastric cancer tumors

Data taken from clinicaltrials.org *. Patient-derived organoids can identify patient-specific drug targets and thus work as a tool for personalized medicine. Links to the studies can be found in Appendix A. * URL last accessed on 20 October 2023.

## Data Availability

The links to data available in Table 3 can be found in Appendix A. Any other information related to the article can be obtained upon request.

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
