# Peer review of "Modeling Liver Development and Disease in a Dish"

_ijms, 2023, doi:10.3390/ijms242115921_

Round 1

Reviewer 1 Report

The manuscript reviews the use of organoids for modelling organ development and disease and how they compare to other commonly used models such as cell lines and animal models and xenografts. With particular focus on the liver, the major applications of these models as a development research tool, disease modelling, regenerative medicine, precision medicine, drug screening are all covered as well as the topical subject of virus infection. The future potential of organ-on-chip models is also described.

The review is well written and effectively covers all the most relevant applications of liver organoids.

There are only minor grammatical errors in the manuscript and should be accepted with no major revisions.

Only some minor grammatical errors.

Author Response

Reviewer-1: The manuscript reviews the use of organoids for modelling organ development and disease and how they compare to other commonly used models such as cell lines and animal models and xenografts. With particular focus on the liver, the major applications of these models as a development research tool, disease modelling, regenerative medicine, precision medicine, drug screening are all covered as well as the topical subject of virus infection. The future potential of organ-on-chip models is also described.

The review is well written and effectively covers all the most relevant applications of liver organoids.

There are only minor grammatical errors in the manuscript and should be accepted with no major revisions.

Response: Thank you for reviewing our article. We have fixed the grammatical mistakes that were overlooked during the writing of this article and hope the article is considered for publication. We thank you for your valuable and constructive review.

Reviewer 2 Report

Waqas Iqbal et al. are presenting a review article on the development of liver disease modeling in a dish. Although the topic is exciting, this reviewer lost immediately his enthusiasm in the second paragraph. The author focuses too much on other organs rather than the liver when explaining the differences between animal and human models. Although accurate, many examples of the differences between animal and human models are not relevant to the proposed review, which should focus only on the liver. The author diverges in the same chapter from the title of the chapter itself, talking about viruses and bacteria, and immortalized cell lines that have nothing to do with the title of the chapter "Animal models and divergences in developmental trajectories", confusing the reader and showing a non-organized layout of the manuscript. In the same chapter, the author starts to talk about other cell culture techniques that do not pertain to animal models, further confusing the reader. This is also true for the other chapters, which in spite of their titles discuss topics not pertinent to the proposed chapter title.

I would suggest the authors perform a thorough review of their manuscript, organize the different chapters in a more logical way, and discuss each topic in the appropriate section, rather than mix everything in the same chapter. Perhaps it could be useful to change the title of the review to a broad topic rather than focus on the liver.

The English language is acceptable.

Author Response

Reviewer-2: Waqas Iqbal et al. are presenting a review article on the development of liver disease modeling in a dish. Although the topic is exciting, this reviewer lost immediately his enthusiasm in the second paragraph. The author focuses too much on other organs rather than the liver when explaining the differences between animal and human models. Although accurate, many examples of the differences between animal and human models are not relevant to the proposed review, which should focus only on the liver. The author diverges in the same chapter from the title of the chapter itself, talking about viruses and bacteria, and immortalized cell lines that have nothing to do with the title of the chapter "Animal models and divergences in developmental trajectories", confusing the reader and showing a non-organized layout of the manuscript. In the same chapter, the author starts to talk about other cell culture techniques that do not pertain to animal models, further confusing the reader. This is also true for the other chapters, which in spite of their titles discuss topics not pertinent to the proposed chapter title.

I would suggest the authors perform a thorough review of their manuscript, organize the different chapters in a more logical way, and discuss each topic in the appropriate section, rather than mix everything in the same chapter. Perhaps it could be useful to change the title of the review to a broad topic rather than focus on the liver.

Response: The author focuses too much on other organs rather than the liver when explaining the differences between animal and human models.

Thank you very much for thoroughly reviewing the article. As per you suggestion we have made some significant changes to the chapter “Animal models and divergences in developmental trajectories”. We tried to make it as specific to our title as possible. The paragraph pertaining to immortalized cells has been removed.

Response: The author diverges in the same chapter from the title of the chapter itself, talking about viruses and bacteria, and immortalized cell lines.

We have removed any portion that might diverge from the main theme of the chapter. In the section about viruses and bacteria in second chapter “Animal models and divergences in developmental trajectories” we tried to explain the shortcomings of animal models in recapitulating human microbial diseases and added some liver specific information as well, such as, HBV and HCV viruses. The chapter gives a general overview of the difference and hence contains information related to organoid models for several organs to fully apprehend the reader about the differences in general.

Response: This is also true for the other chapters, which in spite of their titles discuss topics not pertinent to the proposed chapter title.

Thank you for pointing this out. We have reorganized all the chapters and even added some relevant information where necessary in other chapters as well.

Addition of a paragraph in chapter “Liver organoid models” at line 239 and have revamped the chapter on liver organoids in disease modeling. Rearranged and removed irrelevant sections in the chapter 5. Added liver related information in chapter 7, which is a general overview of organoid validation. “Organoids in viral infection” a sub-section of chapter 8 has been thoroughly improved.

Chapter 10 again discusses the role of organoids in patient specific treatments where we try to give an overview of the topic and discuss liver organoids where appropriate.

We hope the changes we made would be satisfactory and tried to address the shortcomings pointed out by the respected reviewer to the best of our ability. We would like to thank the reviewer again for their suggestions that we believe have improved the overall quality of our article.

Reviewer 3 Report

The manuscript entitled “Modeling Liver Development and Disease in a Dish” reviews comprehensively the research to reproduce the liver on a culture dish to elucidate pathological conditions and utilize it for treatment. Although there are some excessive contents deviating from the title (for instance, part of Tables 1 and 3), the manuscript is substantially worthy of publication in International Journal of Molecular Sciences. Before acceptance, I would like the authors to improve the following points.  

1. Line 26: Please make the section title more informative, such as “1. Introduction - historical overview of organoid research” or something.  

2. Line 59: Please omit the indication of Fig.2. The indication is normally placed in the first section that gives a specific description.  

3. Line 394: Please make the title of Table 1 more descriptive.  

4. Line 758: It would be helpful for readers to link each item in Table 3 to the respective website in ClinicalTrials.gov by supplementary table.

There are some excess spaces. Please check and omit them.

Author Response

Reviewer 3: The manuscript entitled “Modeling Liver Development and Disease in a Dish” reviews comprehensively the research to reproduce the liver on a culture dish to elucidate pathological conditions and utilize it for treatment. Although there are some excessive contents deviating from the title (for instance, part of Tables 1 and 3), the manuscript is substantially worthy of publication in International Journal of Molecular Sciences. Before acceptance, I would like the authors to improve the following points.  

1. Line 26: Please make the section title more informative, such as “1. Introduction - historical overview of organoid research” or something.  

2. Line 59: Please omit the indication of Fig.2. The indication is normally placed in the first section that gives a specific description.  

3. Line 394: Please make the title of Table 1 more descriptive.  

4. Line 758: It would be helpful for readers to link each item in Table 3 to the respective website in ClinicalTrials.gov by supplementary table

Response: Line 26: Please make the section title more informative, such as “1. Introduction - historical overview of organoid research” or something.

Thank you very much for reviewing our article. As per you suggestion we have changed the title the section “Introduction” to “Introduction – historical overview”.

Response: Line 59: Please omit the indication of Fig.2. The indication is normally placed in the first section that gives a specific description.

Thank you for pointing this out. We have moved the indication of Fig.2 to line no. 37.

Response: Line 394: Please make the title of Table 1 more descriptive.

The title of the table have been changed and we also added some relevant information in the legend. Table 1 moved to line 74. We also added abbreviations in each table/Figure that were missing.

Response: Line 758: It would be helpful for readers to link each item in Table 3 to the respective website in ClinicalTrials.gov by supplementary table

A new supplementary table has been added with links to each study mentioned in table 3. We hope the changes we made would be deemed satisfactory. Thank you for your positive and constructive review.

Round 2

Reviewer 2 Report

The authors made substantial modifications to the manuscript and the different paragraphs are more clear. The manuscript's overall quality has improved.